# Annular Erythemas and Purpuras

**DOI:** 10.3390/life13061245

**Published:** 2023-05-24

**Authors:** Nicolas Kluger

**Affiliations:** Department of Dermatology, Allergology and Venereology, Helsinki University Hospital & University of Helsinki, 00250 Helsinki, Finland; nicolas.kluger@hus.fi

**Keywords:** annular dermatosis, erythema annulare, erythema marginatum, erythema migrans, erythema centrifugum, figurate erythema, annular purpura

## Abstract

Annular dermatoses are a heterogeneous and extremely diverse group of skin diseases, which share in common annular, ring-like patterns with centrifugal spreading. Numerous skin diseases can sometimes display annular lesions, but some specific skin conditions are originally annular. We take the opportunity to review here mainly the causes of primary annular erythemas and their differential diagnoses, but also the rare causes of annular purpuras.

## 1. Introduction

Annular dermatoses (AD) are a heterogeneous group of skin diseases, whose common feature is essentially the annular or circular arrangement of the lesions with centrifugal spreading [1]. They belong to the group of figurate dermatoses, to which can be added linear or serpiginous dermatoses, for example. They are a fascinating curiosity for the dermatologist, as they form patterns and arabesques on the patient’s body.

Clinically, AD rarely present as a flat erythematous macule or spot. The active border is often redder and palpable, and the initial lesion is a papule or plaque that will spread as a ring. The confluence of nearby lesions gives a polycyclic appearance. Finally, the rings may be closed circles or ovals, or well opened, arciform or crescent-shaped. In its recent update of the glossary of descriptive terms for skin lesions, the International League of Dermatological Societies (ILDS) suggested the use of the following terms to describe the lesions: annular, arciform, polycyclic and oval [2].

AD are most often annular erythema. Careful examination of the surface of the rash can also determine whether the process arises from the epidermis (dermatophyte) or from the dermis (erythema migrans, etc.). However, purpuras can also display a ring-like pattern in cases of cutaneous vasculitis, pigmented purpura or infection. In this review, we mainly discuss AD from the perspective of erythema annulare, and at the end, we review some cases of purpura annulare. Pediatric erythema annulare is a broad group of dermatoses whose differentiation is often based on subtle clinical and histologic nuances [3]. We discuss only some of these pediatric entities.

## 2. Classification of Annular Dermatoses

AD can be classified in different ways. (i) AD can be either primarily or secondarily annular. In the first case, the annular pattern is an intrinsic characteristic of the dermatoses and defines them (erythema annulare migrans, centrifugal erythema annulare of Darier, tinea, granuloma annulare, subacute lupus, etc.). In the second case, a large number of skin conditions can display any annular pattern among their possible clinical presentation. Most likely, any dermatologic disease may display an annular pattern, so the list is long, as follows: mycosis fungoides, syphilis, roseola, seborrheic dermatosis, sarcoidosis, herpetic dermatitis, linear IgA dermatosis, etc. (Table 1, Figure 1). (ii) AD can be classified according to their causes (infectious, para-infectious, inflammatory, paraneoplastic, drug-induced) [4], (iii) according to the age of onset (infant/adult, Table 2) [3], (iv) according to the type of histological infiltrate (Table 3) [4], or finally, (v) according to the acute or chronic onset. This last classification [1] seems to us to be the simplest in daily practice. Indeed, classification by group of causes [4] lead to a large laundry list mixing common and rare conditions, as well as acute and chronic conditions. Histopathological classification [5] is of interest when a biopsy is taken to allow a clinical confrontation; however, this is not always necessary.

## 3. Acute Annulare Dermatoses

### 3.1. Pityriasis Rosea

Pityriasis rosea is a benign squamous eruptive exanthema due to HHV-6 and HHV-7 that mainly affects the young. The classic initial erythemato-squamous, round or oval, annular “medallion” or herald patch with raised edges and a clear center, and of variable size, precedes the generalized eruption [6]. Its identification allows the diagnosis (Figure 2). On the other hand, if the primary lesion is seen alone before the rash, tinea infection may be wrongly suspected. Differential diagnoses include secondary syphilis and psoriasis.

### 3.2. Erythema Marginatum

Erythema marginatum was first described in 1831 by Bright and has been referred to by several names (érythème marginé discoïde de Besnier, Lehndorff–Leiner’s erythema annulare rheumaticum, Keil’s erythema marginatum rheumaticum). Although a rare manifestation (<6% of cases), erythema marginatum is still considered a specific lesion of post-streptococcal rheumatic fever (RF) and a major criterion in the new 2015 revision of the Jones criteria [7]. RF and erythema marginatum usually affect children, and more rarely, adults [8]. Erythema marginatum is characterized by a painless and non-pruritic, annular, polycyclic, raised or non-raised border rash that is red to purplish in color and usually rapidly evolving [9] with an occasional disappearance and reappearance of lesions. Centrifugal evolution is not systematic [8]. Erythema marginatum affects the trunk, extremities and sometimes the face. It occurs after untreated angina with fever in association with migratory inflammatory joint pain and an inflammatory syndrome. In case of a heart murmur, an ultrasound must be performed to eliminate a valvulopathy. A skin biopsy shows a perivascular neutrophilic inflammatory infiltrate [9], that is sometimes lymphocytic [8] in the dermis, without vasculitis and without alteration of the epidermis [8,9]. Direct immunofluorescence is negative. The evolution is favorable under antibiotic therapy with penicillin A in the following weeks. A rapid angina test is positive [8], and antistreptolysins can be positive [9]. The infection is not immunizing, and the rash may recur in case of a new infection [8]. RF has become rare in developed countries due to the use of antibiotics for streptococcal infections. Given the decrease in the prevalence of RF, it is important to be aware of this diagnosis in case of fever and joint pain after angina, especially in patients from developing countries.

### 3.3. Erythema Marginatum with Hereditary Angioedema

The occurrence of a non-pruritic erythema annulare, reticular or serpiginous would concern 56% of patients with hereditary angioedema [10]. This “erythema marginatum” has been observed since the end of the 19th century. It affects the upper part of the trunk and the limbs. It may precede or accompany angioedema attacks [10,11]. If it is mistaken for urticaria, the diagnosis of hereditary angioedema is delayed. Its pathophysiology is unclear but may involve bradykinin [10].

### 3.4. African Trypanosomiasis

The lymphatic blood phase of trypanosome dissemination occurs 1 to 3 weeks after the infesting tsetse fly bite. It associates fever, hepatosplenomegaly, cervical adenopathies and trypanides. These are erythematopapular lesions of variable size which may take on the appearance of a polycytic centrifugal annular erythema with a marked erythematous active border, most often on the trunk or the roots of the limbs. In a suggestive context (expatriate living in black Africa), these trypanids are characteristic but rare (10 to 50% of cases), fleeting, recurrent, leave no sequelae and may naturally go unnoticed on dark skin [12,13,14].

## 4. Chronic Annular Dermatoses

### 4.1. Tinea Corporis

Infection of the glabrous skin by *Trichophyton rubrum* or *T mentagrophytes* constitutes the prototype of ring dermatosis. It initially presents as a small pinkish macule of a progressive centrifugal extension forming a circle or a closed oval, with an active vesicular border, sometimes scaly, more or less pruritic, until it ends up in a large placard, sometimes polycyclic, in case of confluence with other lesions. The center of the lesion takes on a brown hue (Figure 3). Fold dermatophyte has a similar appearance as an extensive, unilateral, ring-shaped placard with an active border and a brown center. The application of local corticosteroids can significantly delay the diagnosis by masking the initial appearance (tinea incognito). However, the annular appearance may remain visible and suggestive (Figure 4). Finally, tinea imbricata or tokelau is a chronic superficial dermatophyte due to *T concentricum* (anthropophilic germ) endemic to the South Pacific islands, but also to South Asia and South America. It is responsible for particularly impressive concentric or annular figured skin lesions [15,16].

### 4.2. Erythema Migrans

Erythema migrans (EM) is the first manifestation of *Borrelia burgdorferi* infection and indicates Lyme disease. It occurs in 90% of patients between 2 days and 3 months after a tick bite; on average, it occurs 2 weeks after the bite [17]. The bite of an Ixodes nymph may go unnoticed. The typical form of EM is an erythematous macule or papule with a centrifugal annular extension and central clearing. It is highly variable in size [18]. It may be asymptomatic, pruritic or painful. If the central site of the inoculation is still visible, the EM displays the typical bull’s eye appearance (Figure 5).

However, in practice, EM can be highly polymorphous, making borreliosis a major simulacrum in the same way that syphilis is in endemic areas (Figure 6) [18,19].

General signs are sometimes present. Serology is neither useful nor recommended at that stage of the disease. In the case of an atypical form, a biopsy reveals a deep and superficial perivascular and interstitial lymphocytic infiltrate with low neutrophil counts. The presence of plasma cells is suggestive of the diagnosis. Warthin–Starry staining may reveal the spirochete in the papillary dermis, but currently, PCR techniques confirm the presence of Borrelia in the skin [20]. Empirical treatment is based on amoxicillin or doxycycline per os for two weeks. In the absence of treatment, EM lesions disappear rapidly with a risk of secondary dissemination. The presence of multiple, smaller, non-expanding annular lesions at a distance from the site of the inoculation indicates a hematogenous and lymphatic dissemination of the disease and is common in the European form of borreliosis [17,18].

### 4.3. Granuloma Annulare

Granuloma annulare (GA) is a benign non-infectious granulomatous skin disorder of unknown origin. GA has been associated with a wide number of conditions such as diabetes, HIV infection, malignancy or medications [21]. There are various clinical variants of GA. The most common form of GA presents as a pink or flesh-colored non-squamous annular erythematous plaque with an active border with a palisading or interstitial granulomatous inflammatory infiltrate (Figure 7). It can be localized to the extremities or generalized [22]. GA is often self-limited and does not require treatment. The generalized form displays a protracted evolution and is more difficult to treat. The list of treatments that have been given for GA is exceedingly long [21].

### 4.4. Subacute Cutaneous Lupus Erythematosus

Subacute cutaneous lupus erythematosus (SCLE) most commonly affects white women over 50 years of age. The lesions are polycyclic, with an erythemato-squamous or vesiculous and crusty border and a greyish hypopigmented center that is sometimes covered with telangiectasias. They regress, leaving post-inflammatory hypo- or hyperpigmentation and telangiectasia, but without atrophy. The lesions predominantly occur on photoexposed areas (face, neck, décolleté, shoulders, extension side of the arms and back of the hands) and on the trunk (Figure 8).

Hyperkeratosis, mucosal body atrophy, extensive basal keratinocyte degeneration, thickening of the basement membrane and the presence of a dermal CD4 lymphocytic infiltrate are suggestive of the diagnosis. Direct immunofluorescence is inconsistently positive [23]. SCLE is the most frequently reported form of cutaneous drug-induced lupus erythematosus. It should always be considered in case of onset among the elderly patients. The clinical presentation, patterns and distribution of the cutaneous lesions are not distinguishable from idiopathic SCLE. Culprit drugs include terbinafine, anti-TNF alpha, antieplieptics and proton pump inhibitors [24]. Treatment includes photoprotection, the application of dermocorticoids and the initiation of hydroxycholoroquine. Systemic involvement should be ruled out [23].

### 4.5. Neonatal Lupus

The transplacental passage of maternal anti-Ro/SSa and/or anti-La/SSb or anti-U1-RNP antibodies may be accompanied by skin lesions that occur early in the first three months of life and resolve during the first year after the elimination of maternal antibodies. The rash is similar to adult subacute lupus with polycyclic annular lesions. Residual pigmentation may occur. Atrophy or scarring is rare. Erythema gyratum atrophicans transiens neonatale of Giannotti and Ermacora [25] is probably a variant with central whitish atrophy [26].

### 4.6. Erythema Annulare with Ro-SSA Antibodies and Sjögren’s Syndrome

Lesions of erythema annulare, similar to those seen in SCLE, were also reported in patients with Sjögren’s syndrome [27]. In addition, a “borderline” sub-entity was reported by Japanese authors in the form of a deep annular erythema with thick borders and a clear doughnut center, mainly affecting the face, and sometimes the upper limbs. The histology shows a perivascular CD4 lymphocytic infiltrate and edema of the dermis without vasculitis or epidermal involvement. The appearance is evocative of lupus tumidus. Patients always present with Ro/SSa antibodies, but the diagnosis may be uncertain between lupus and Sjögren’s syndrome [28].

### 4.7. Neutrophilic Dermatoses

Neutrophilic dermatoses are a group of rare skin conditions characterized by the presence of an abnormally high number of neutrophils in the epidermis and/or the dermis and/or the subcutis. These conditions can present with a variety of clinical features such as painful erythematous (red) papules, plaques, pustules and vesicles. Some of the most common neutrophilic dermatoses are Sweet syndrome and pyoderma gangrenosum [29]. Lesions of Sweet syndrome can sometimes evolve secondarily into polycyclic annular lesions (Table 3). However, there are also some conditions among the neutrophilic dermatoses (ND) group, during which the lesions are primarily annular.

Neutrophilic erythema multiforme of infancy is an exceptional inflammatory dermatosis affecting healthy newborns or small children with no other underlying pathology. It is characterized by asymptomatic, annular erythematous lesions, sometimes polycyclic, without vesicles or scaling. The histology shows a neutrophilic infiltrate and leukocytoclasia, without vasculitis. The dermatosis may progress to remission or chronicity [3].

Chronic recurrent annular neutrophilic dermatosis (CRAND) was first described in 1989. CRAND is a peculiar and rarely described form of neutrophilic dermatosis marked by an annular and chronic course, and a histological involvement like Sweet syndrome, but without any general signs, biological abnormalities or underlying systemic pathology [30]. It seems to affect adult women after 40 years of age. Oral steroids, colchicine and dapsone are the treatments of choice [31]. Croci-Torti et al. discuss some clinical cases in the literature that do not yet fit into specific nosological entity [31] cases as erythema annulare with a neutrophilic infiltrate on histology [32].

### 4.8. (Darier’s) Erythema Annulare Centrifugum

Erythema annulare centrifugum (EAC) is characterized by mildly symptomatic annular plaques on the trunk, buttocks and proximal parts of the limbs, while it spares the face and extremities. EAC can occur at any age, with a predominance in people who are in their 50s [33]. The number of lesions is also variable, averaging from two to five. In his original description, Darier stated that pink rings consist of a peripheral bump, a 3–5 mm cord, with a gentle slope to the skin on the inner side of the ring and a sharp edge toward the healthy skin [1]. Scaling of the inner edge of the ring is sometimes noted, while the central area takes on a bister or buff color (Figure 9).

The lesions may coalesce into polycyclic rings and evolve over several days, weeks or months. The following two forms are currently considered: the superficial form and a deep form (that has been described by Darier). A biopsy of recent lesions of superficial CAE shows a superficial spongiform dermatitis with an eosinophilic infiltrate and sometimes neutrophils. In older lesions, the infiltrate is lymphocytic without spongiosis. In the deep form, there is a superficial and deep perivascular lymphocytic infiltrate. In this case, various differential diagnoses arise, such as lupus tumidus or erythema migrans. The etiology and pathophysiology of EAC are not known. Treatment is only proposed in the case of discomfort. The disease resolves spontaneously within a year, although prolonged courses are possible.

### 4.9. Eosinophilic Annular Erythema

Eosinophilic annular erythema (EAE) is a rare entity in the literature. It was first described in 1981 by Peterson and Jarrat [34]. The first cases were described in children as annular erythema of infancy [34,35], but later, similar cases were observed in adults [35]. It is a chronic dermatosis, characterized by arciform annular plaques with a dark red center, which may affect the trunk, limbs or face. The edges are raised, smooth, non-scaly, pruritic [35] or not pruritic [36]. The lesions disappear without atrophy or scarring [35], but new lesions may reappear on the top of regressing lesions [36]. The general condition is preserved. Arthralgias may occur. Usually, no systemic disease is reported, although we found one case associated with a chronic hypereosinophilic syndrome [37] and one associated with metastatic prostate cancer [38]. The evolution is variable, ranging from spontaneous resolution in a few weeks without recurrence [39], to chronic forms over several years. The biopsy discloses an interstitial, perivascular and perisudoral dermal infiltrate, predominantly lymphocytic and eosinophilic, but poor in neutrophils and without plasma cells. There is no evidence of flame figure, granuloma or vasculitis. Mucin deposits may be present. Vacuolation of the basement membrane and spongiosis and exocytosis in the epidermis are also noted [36]. Circulating hypereosinophilia is inconsistent. Treatment is difficult. Anti-inflammatory drugs such as indomethacin, anti-malarial drugs and even Disulone seem to be the most effective [40]. The question of the link between EAE and Wells syndrome remains open. The distinction is based on clinico-biological differences (no pseudo-cellulitis appearance, absence of hypereosinophilia), histological features (absence of flame figure, granuloma and eosinophilic degranulation) and possibly therapeutic nuances (corticosteroid therapy versus hydroxychloroquine and NSAIDs).

### 4.10. Erythema Gyratum Repens

Described in 1952, erythema gyratum repens is a rare (70–80% of cases) paraneoplastic syndrome whose diagnosis is clinical. It affects the trunk and extremities with characteristic erythematous, raised, scaly, serpiginous, concentric or annular bands arranged in parallel giving a “wood-ribbed” appearance and centrifugal extension. It is mainly associated with bronchial, breast and esophageal cancers. The histology is non-specific with a superficial lymphocytic perivascular infiltrate. The main purpose of the histology is to eliminate differential diagnoses [41]. Cases have been recently associated with COVID-19 infection [42,43] and SARS-CoV-2 vaccination [44].

## 5. Annular Purpura

### 5.1. Majocchi’s Purpura Annularis Telangiectodes

Pigmented purpuric dermatoses are a group of skin disorders characterized by a chronic or persistent course, varied clinical picture (macules, papules and plaques) and unknown origin (e.g., contact allergy, pressure, physical training, medication) [45]. Among this group of pigmented purpuric dermatoses, Majocchi’s purpura annulare (purpura annularis telangiectodes) presents as purpuric patches of a centrifugal extension with a bright red border and a yellowish central area that is sometimes atrophic, located on the lower limbs (Figure 10). It tends to affect young people of both sexes. The rings are a few millimeters thick and covered with telangiectasias and punctiform purpura. The rings are closed or open, isolated or confluent and polycyclic. The size of the lesions is variable. The lesions regress in a few months or years without scarring. Histologically, there is lymphocytic capillaritis with an extravasation of red blood cells and lymphocytic infiltrate without vasculitis [46,47].

### 5.2. Annular Leucocytoclastic Cutaneous Vasculitis

It is not unusual to see an annular pattern within purpuric lesions in cutaneous vasculitis (Figure 11). However, a singular entity was described in 1996 [48] as the skin lesions were purpuric annular plaques of a centrifugal extension with a polycyclic extension to the limbs and trunk, leaving a residual pigmentation. There are no extracutaneous symptoms or impairment of general condition. The histology was consistent with leucocytoclastic vasculitis with inflammatory infiltrate. The lesions regress with dapsone [48], but spontaneous regression is possible [49].

### 5.3. Post-Infectious Purpura

A few curiosities have been reported, such as pityriasis rosea in a purpuric and annular variant [50] or annular hemorrhagic purpura in hantavirus, as we reported in a case of Puumala virus infection in Finland [51].

## 6. Conclusions

Annular dermatoses encompass a large number of various and heterogenous conditions that make it difficult to define decisional algorithms. Diagnoses can be only achieved via proper history intake, past dermatological diseases, proper analysis of the clinical presentation (acute/chronic, isolated/multiple lesions, erythema/purpura, etc.) and microscopic findings. Annularity is a dynamic process, and the lesions may change with time and between observations. Physicians should also enquire whether the patient has taken pictures of the skin lesions at the very beginning of their occurrence or at various stages of their evolution. Differential diagnoses of annular dermatoses encompass, by definition, the other annular dermatoses.

## Figures and Tables

**Figure 1 life-13-01245-f001:**
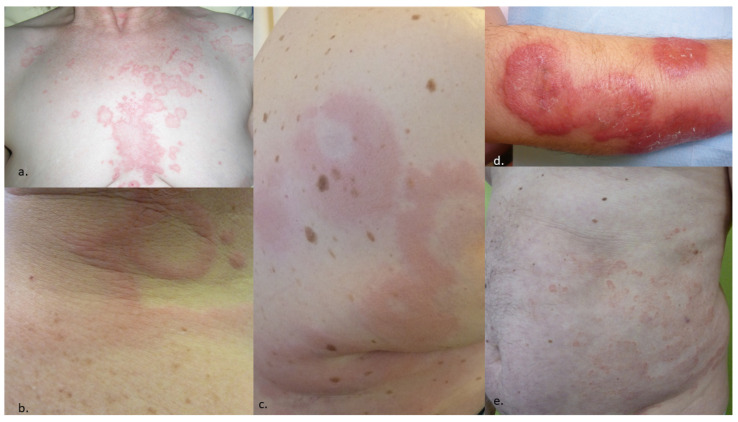
Examples of dermatoses with possible presentation as annular lesions. (**a**) Seborrheic dermatitis of the trunk, (**b**) acute urticaria, (**c**) urticarial vasculitis, (**d**) Sweet syndrome, (**e**) chronic eczema.

**Figure 2 life-13-01245-f002:**
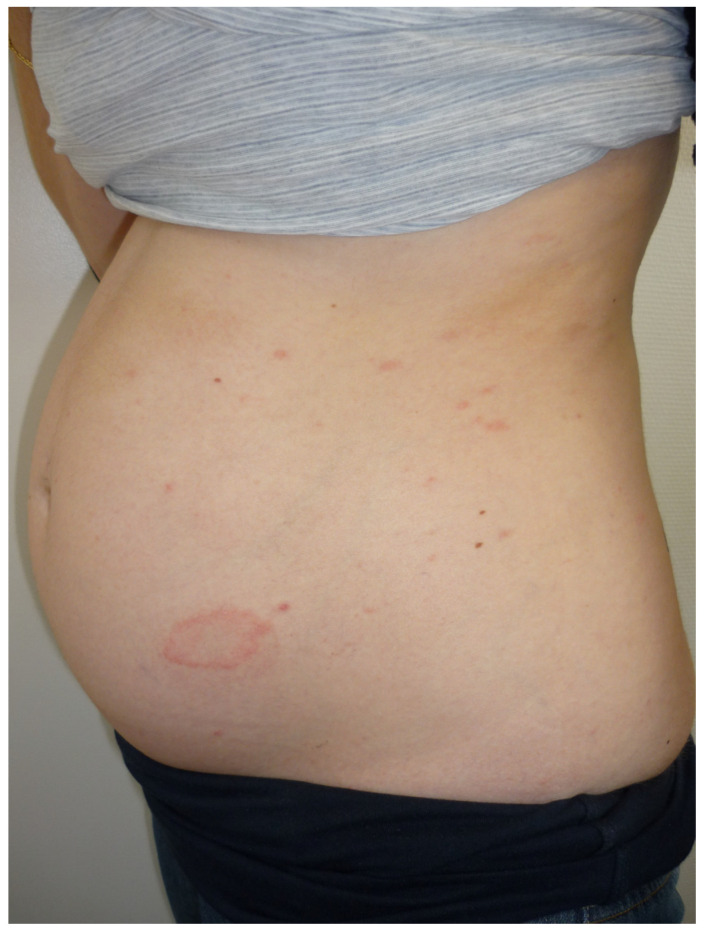
Pityriasis rosea in a pregnant woman with its typical initial ring medallion.

**Figure 3 life-13-01245-f003:**
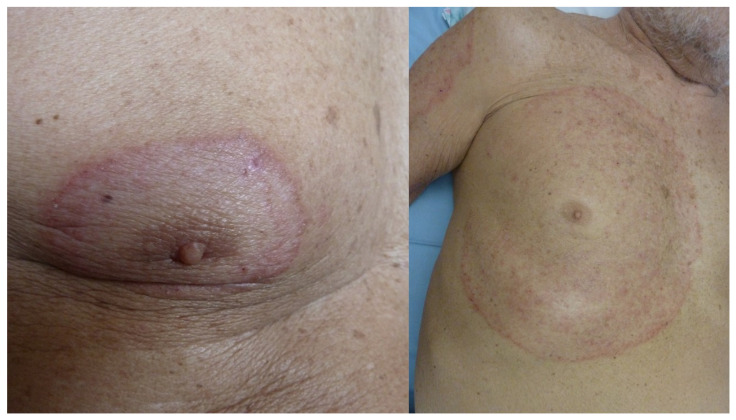
Dermatophytosis of the glabrous skin with *T. rubrum* centered on the right nipple (**left**). Extension and progression over 9 months due to poor compliance with home treatment (**right**).

**Figure 4 life-13-01245-f004:**
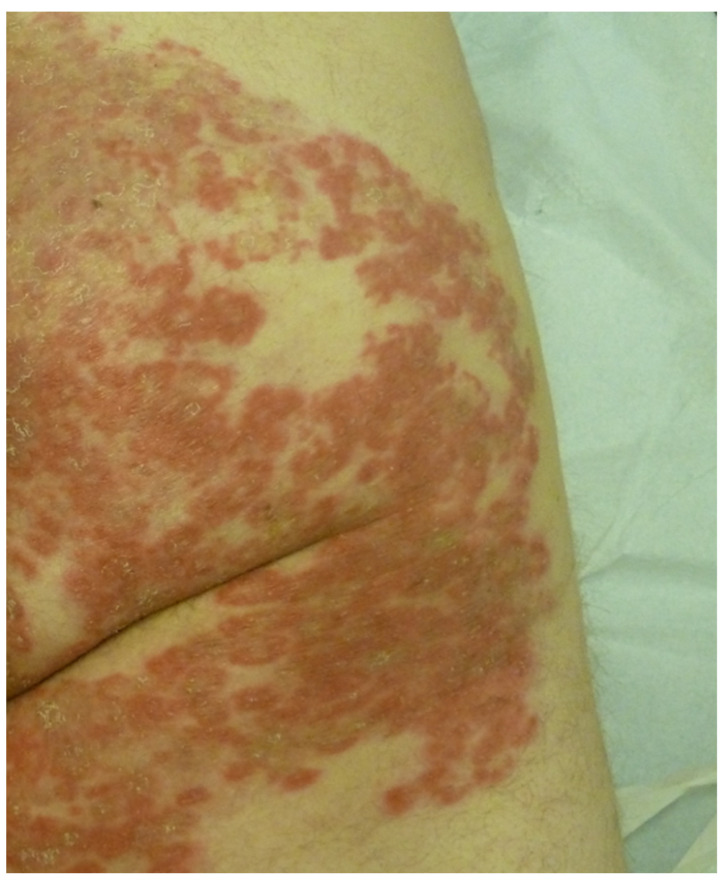
Close-up view of tinea corporis incognito initially treated with very high potency dermocorticoids.

**Figure 5 life-13-01245-f005:**
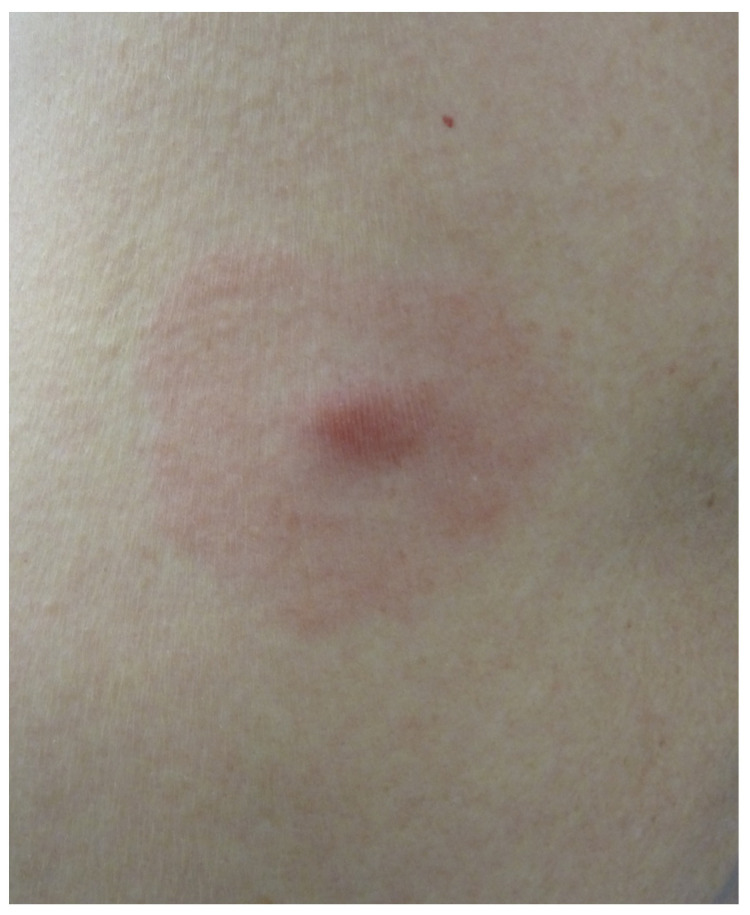
Erythema migrans with a bull’s eye sign.

**Figure 6 life-13-01245-f006:**
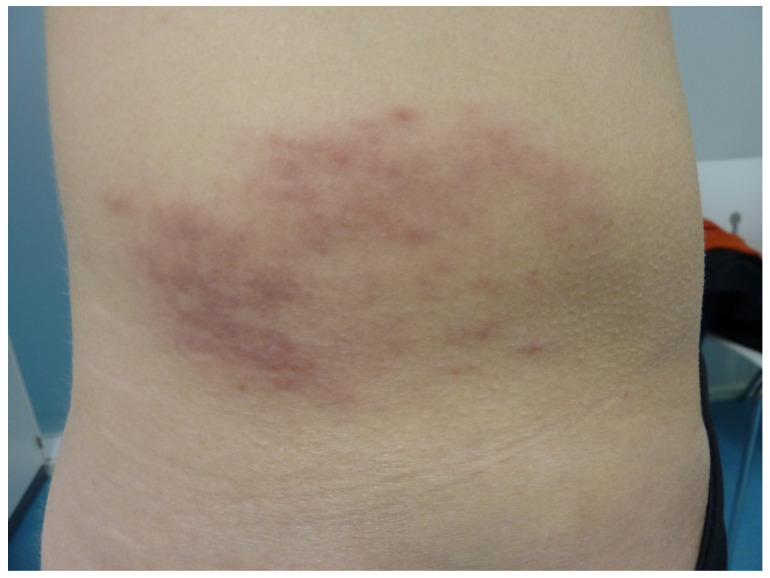
Atypical erythema migrans presenting an annular hematoma.

**Figure 7 life-13-01245-f007:**
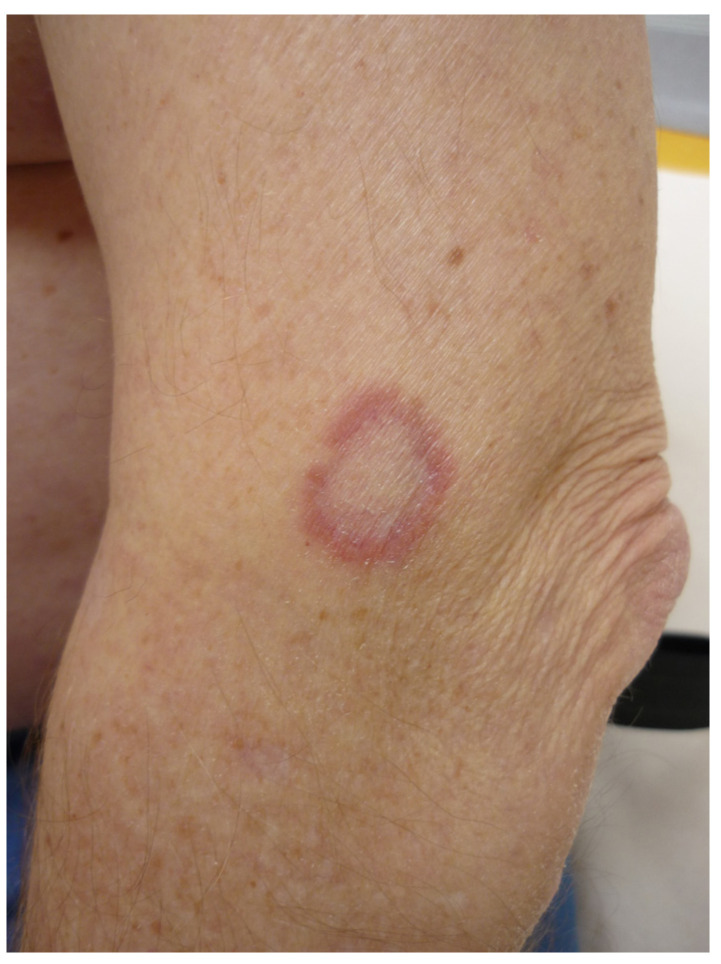
Granuloma annulare of the elbow.

**Figure 8 life-13-01245-f008:**
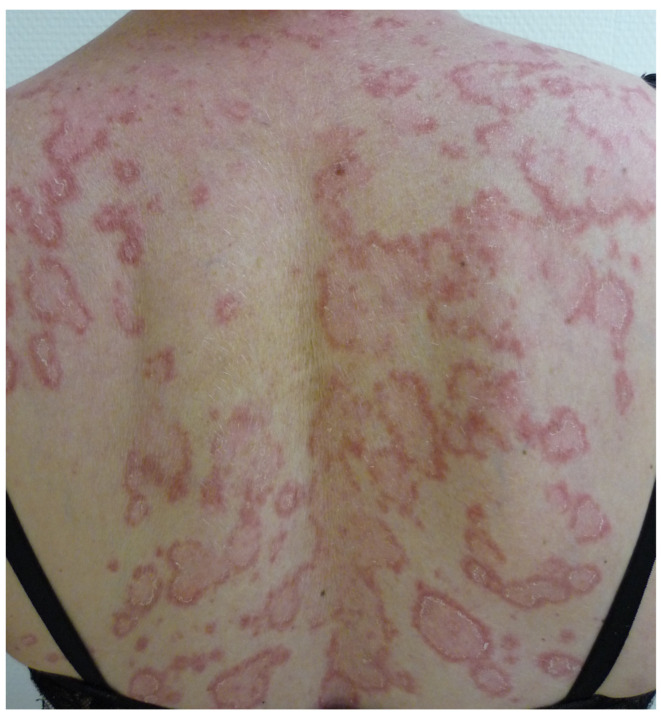
Subacute cutaneous lupus erythematosus of the back.

**Figure 9 life-13-01245-f009:**
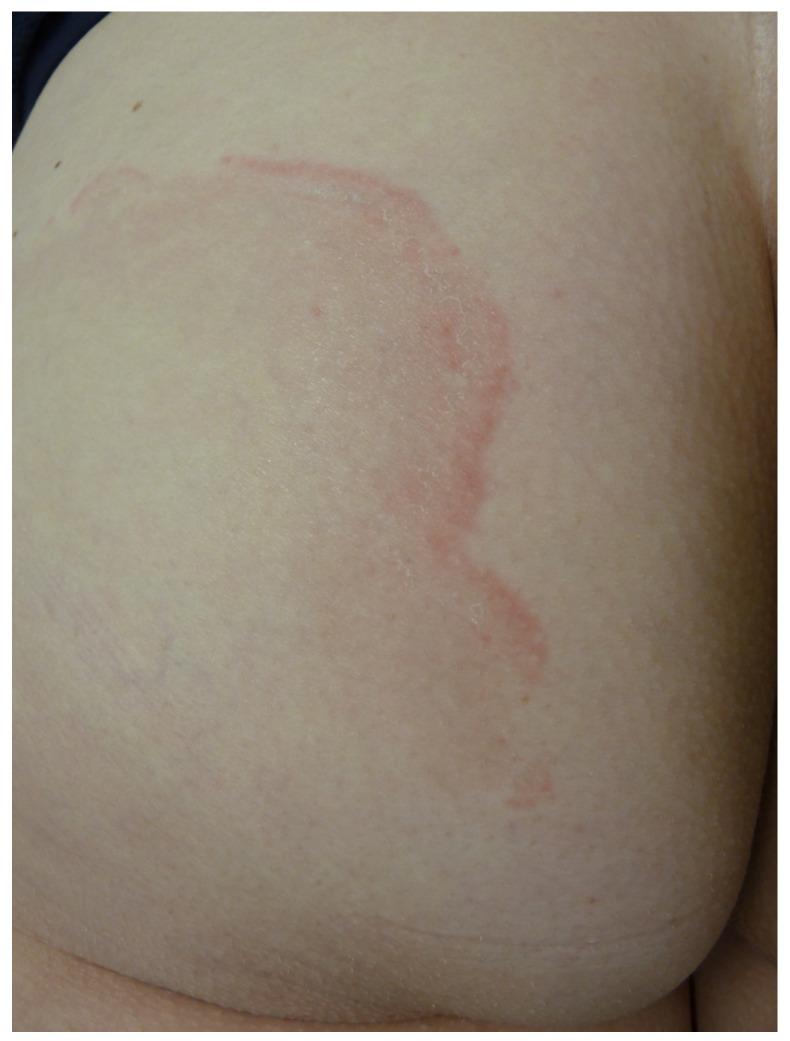
Erythema annulare centrifugum.

**Figure 10 life-13-01245-f010:**
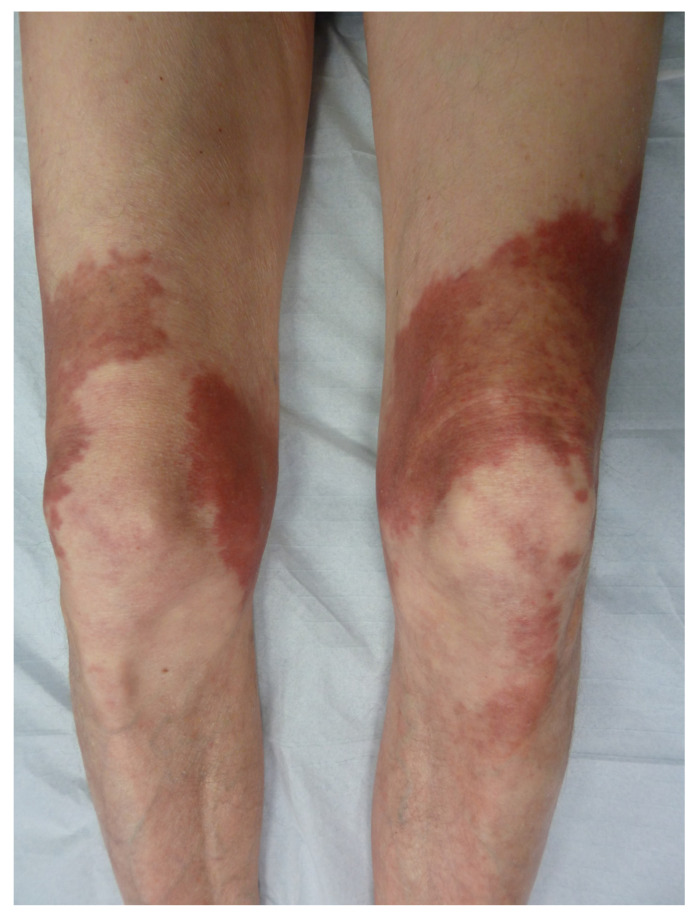
Majocchi’s purpura annulare.

**Figure 11 life-13-01245-f011:**
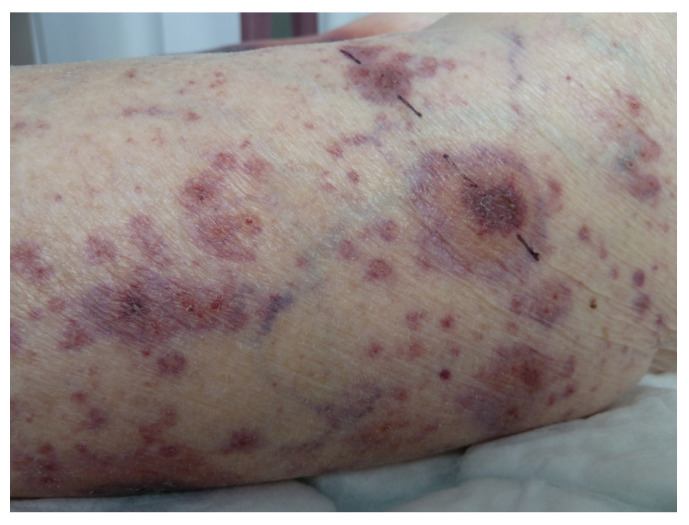
Cutaneous vasculitis disclosing annulare features.

**Table 1 life-13-01245-t001:** Main dermatoses that can *sometimes* display an annular presentation (non-exhaustive list).

Seborrheic dermatosis
Urticaria and urticarial vasculitis
Sweet syndrome
Eosinophilic dermatosis, Wells cellulitis
Psoriasis: annulare psoriasis, psoriasis gyrata
Eczema
Leprosy
Granuloma faciale
Sarcoidosis
Mycosis fungoides

**Table 2 life-13-01245-t002:** Annular erythema of infancy ^a^ (modified from Patrizi et al. [3]).

Neonatal lupus erythematosus variant: erythema gyratum atrophicans transiens neonatale Erythema annulare with anti-Ro/SSa antibodies (Sjögren) Erythema marginatum rheumaticum Erythema marginatum associated with hereditary angioedema Erythema migrans Erythema annulare centrifugum variant: familial annular erythema Annulare erythema of infancy variant: neutrophilic figurated erythema Eosinophilic annular erythema

^a^ Common dermatoses are not included (psoriasis, tinea, etc.).

**Table 3 life-13-01245-t003:** Annular erythema according to the main inflammatory infiltrate (modified from Ríos-Martín et al. [5]).

Lymphocytes	Neutrophils, Eosinophils	Granulomas	Plasmocytes
Erythema annulare with anti-Ro/SSa antibodies	CRAND Eosinophilic dermatosis Eosinophilic erythema annulare Erythema marginatum Neutrophic dermatosis IgA pemphigus Linear IgA dermatosis Neutrophilic erythema annulare of infancy Psoriasis Sneddon–Wilkinson syndrome Urticarial vasculitis	Granuloma annulare Leprosy Sarcoïdose	Erythema migrans Syphilis
Erythema annulare centrifugum			
Erythema gyratum repens			
Erythema migrans			
Erythema multiformis			
Erythrokeratodermia variabilis			
Leprosy			
Lupus			
Mycosis fungoides			

CRAND: Chronic recurrent annular neutrophilic dermatosis.

## Data Availability

Data are unavailable; no new data were generated due to privacy.

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
