# Peer review of "Annular Erythemas and Purpuras"

_life, 2023, doi:10.3390/life13061245_

Round 1

Reviewer 1 Report

The idea of performing a review on annular dermatoses is good but it has not been made in a scientific way and the therefore the review is not exhaustive. This should ne acknowledged in thr conclusion.

I prefer the etiological classification (see Narayanasetty, Indian J Dermatol). Your table 1 (not complete and of poor significance) could be replaced with table 1 of this manuscript.

another classification could be according to the depth (i.e. epidermal, like tinea, dermal like scle, subcutaneous like annular granuloma).

Some more complete references:

- about tokelau, there is this review: Veraldi S, et al. Mycopathologia 2015

- about purpuric dermatoses: Spigariolo CB, J Clin Med 2021

Author Response

We thank the reviewer 1 for the comments

The corrections have been implemented in red to improve readability.

We added the references that were suggested by the reviewer. 

The chapter of annular dermatoses is exceedingly complicated because virtually any dermatosis can be annular. Reporting a long exhaustive laundry list of cutaneous diseases for a non-dermatologist will be rather confusing. The author of this article wishes to give his own point of view of annular dermatoses, by focusing on lesions that are acute versus chronic The reviewer suggested that to copy the table of an another author. The table uses a classification by causes as mentioned in the present manuscript, but is rather a laundry list associating frequent and rare conditions. The reference has been added in the manuscript.

Reviewer 2 Report

The revision includes the main Annular erythemas and purpuras. It’s important to have a discussion before the conclusion, where the main keys of differential diagnosis are discussed, in this way it is a descriptive list of pathologies found in the literature. Furthermore, the conclusion is too short and should also be changed after writing the discussion.

Line 58-63 : no literary mention on the definition of Pityriasis Rosea

line 201-202: some more reference to therapy should be added, such as: doi: 10.3390/pharmaceutics14020294.

Author Response

We thank reviewer 2 for the comments

The article is mainly the description of various conditions that are heterogenous so it is hard to provide a discussion that would unify all of the diseases

Line 58-63 : no literary mention on the definition of Pityriasis Rosea: a paragraph has been developed.

Reviewer 3 Report

In this review, the author mainly discusses annular dermatoses from the perspective of annular erythema, and also includes some cases of annular purpura. 

The article first proposes five classification methods for annular skin diseases: primary or secondary/etiology/onset age/histological infiltration type/acute or chronic. Then, the article focuses on listing various diseases under acute/chronic annular dermatoses classification, and describes their etiology, susceptible population, typical manifestations, pathological manifestations, diagnostic methods, differential diagnosis and treatment methods. Finally, the author discusses several types of annular purpura and their characteristics. The author concludes that in the case of annular dermatoses, the diverse range of conditions and variable presentations may not fit neatly into a single algorithm. Clinicians must consider various factors and perform a thorough evaluation to diagnose and treat these conditions accurately.

The reviewer proposed some suggestions:

Major

1.    It’d be better to emphasize the significance of this review.

2.    List some annular dermatoses classified by their causes.

3.    Add more detailed information on granuloma annulare.

Minor

1.    Fig3. A more detailed description of the difference between the left and right images is needed.

2.    (Fig 6) was written as (Fig 5)

Author Response

We thank the reviewer 3 for the comments

List some annular dermatoses classified by their causes: this is difficult because some annular diseases may difference causes such as Granuloma annulare or Sweet syndrome. 

Add more detailed information on granuloma annulare.

The paragraph about Granuloma annulare has been developed

Fig3. A more detailed description of the difference between the left and right images is needed: Done 

(Fig 6) was written as (Fig 5): corrected

Reviewer 4 Report

The authors have submitted the manuscript entitled: '' Annular erythemas and purpuras''.  The manuscript is well written and the topic is very usable for clinical practice. For educational purposes, it would be appropriate for each treated pathology:  1) to insert the images where missing;  2) to include some possible differential diagnoses, like you have already done, for example, in the case of Pityriasis rosea.

Author Response

For educational purposes, it would be appropriate for each treated pathology:  1) to insert the images where missing.  Unfortunately this is not possible because some conditions are exceedingly rare. the readers can report for further iconography to other articles or internet   2) to include some possible differential diagnoses, like you have already done, for example, in the case of Pityriasis rosea. The differential diagnoses of annulare dermatoses are other annular dermatoses so lists of differential diagnoses for every condition would be rather confusing for the readers.

Round 2

Reviewer 1 Report

The paper has been improved